# Congenital Myopathy as a Phenotypic Expression of *CACNA1S* Gene Mutation: Case Report and Systematic Review of the Literature

**DOI:** 10.3390/genes14071363

**Published:** 2023-06-28

**Authors:** Gemma Marinella, Alessandro Orsini, Massimo Scacciati, Elisa Costa, Andrea Santangelo, Guja Astrea, Silvia Frosini, Rosa Pasquariello, Anna Rubegni, Giada Sgherri, Martina Corsi, Alice Bonuccelli, Roberta Battini

**Affiliations:** 1Department of Neuroscience, IRCCS Stella Maris Foundation, 56128 Pisa, Italy; gemma.marinella@fsm.unipi.it (G.M.); guja.astrea@fsm.unipi.it (G.A.); silvia.frosini@fsm.unipi.it (S.F.); rosa.pasquariello@fsm.unipi.it (R.P.); anna.rubegni@fsm.unipi.it (A.R.); giada.sgherri@fsm.unipi.it (G.S.); 2Pediatric Neurology, Pediatric University Department, Azienda Ospedaliera Universitaria Pisana, University of Pisa, 56100 Pisa, Italy; aorsinimd@gmail.com (A.O.); scacciatimassimo@gmail.com (M.S.); androsantangelo@gmail.com (A.S.); al.bonuccelli@ao-pisa.toscana.it (A.B.); 3Department of Clinical and Experimental Medicine, University of Pisa, 56126 Pisa, Italy; 4Department of Preventive and Occupational Medicine, Azienda Ospedaliera Universitaria Pisana, University of Pisa, 56126 Pisa, Italy; dott.martinacorsi@gmail.com

**Keywords:** congenital myopathies, *CACNA1S*, dihydropyridine receptor congenital myopathy, CACNA1S myopathy

## Abstract

Background: Congenital myopathies are a group of clinically, genetically, and histologically heterogeneous diseases caused by mutations in a large group of genes. One of these is *CACNA1S*, which is recognized as the cause of Dihydropyridine Receptor Congenital Myopathy. Methods: To better characterize the phenotypic spectrum of *CACNA1S* myopathy, we conducted a systematic review of cases in the literature through three electronic databases following the PRISMA guidelines. We selected nine articles describing 23 patients with heterozygous, homozygous, or compound heterozygous mutations in *CACNA1S* and we added one patient with a compound heterozygous mutation in *CACNA1S* (c.1394-2A>G; c.1724T>C, p.L575P) followed at our Institute. We collected clinical and genetic data, muscle biopsies, and muscle MRIs when available. Results: The phenotype of this myopathy is heterogeneous, ranging from more severe forms with a lethal early onset and mild–moderate forms with a better clinical course. Conclusions: Our patient presented a phenotype compatible with the mild–moderate form, although she presented peculiar features such as a short stature, myopia, mild sensorineural hearing loss, psychiatric symptoms, and posterior-anterior impairment gradient on thigh muscle MRI.

## 1. Introduction

Congenital myopathies are a group of clinically, genetically, and histologically heterogeneous diseases characterized by congenital or early-onset muscle weakness of varying degrees and a static or slowly progressive clinical course [1,2]. Many of these congenital myopathies are due to mutations in more than one gene, and mutations in the same gene may lead to a different penetrance and, consequently, a various expressivity. [3]. The increasing use of new technology for exome and genome sequencing has led to the discovery of new disease genes; however, many have still not been identified. For example, recently, both recessive and dominant *CACNA1S* mutations have been recognized to cause Dihydropyridine Receptor Congenital Myopathy [4,5,6].

The *CACNA1S* gene encodes the α-1s subunit of the dihydropyridine receptor (DHPR), a voltage-gated calcium channel and voltage sensor for calcium release in skeletal muscle [7]. The gene spans 90 kb and contains 44 exons. The voltage-dependent L-type calcium channel subunit α-1S (Cav1.1) is organized into four homologous domains (I–IV) and contains five hydrophobic transmembrane segments (S1, S2, S3, S5, and S6) and one positively charged transmembrane segment (S4), which senses changes in the membrane electric field to initiate conformational changes in the protein that both trigger intracellular Ca^2+^ release and promote pore opening. The intracellular “loop” connecting the second and third domains (“II–III loop”) contains a critical region of amino acids (620–764), which is required to mechanically couple changes in the membrane voltage to intracellular Ca^2+^ release via type 1 ryanodine receptor (RYR1) Ca^2+^ release channels located in the terminal cisternae of the sarcoplasmic reticulum (orthograde signaling). The critical domain of the II-III loop also promotes a signal from RYR1 channels that augment DHPR channel activity (retrograde signaling) [8,9,10]. 

Heterozygous, dominant-acting *CACNA1S* mutations have previously been associated with susceptibility malignant hyperthermia triggered by volatile anesthetics (MHS5; MIM#601887) [11,12], hypokalemic periodic paralysis (HOKPP; MIM#170400), which is characterized by acute flaccid weakness episodes associated with low serum potassium levels (<2.5 mEq/L) and, occasionally, the development of permanent muscle weakness after several years of continuous episodes [13,14], and thyrotoxic periodic paralysis (TTPP1; MIM#188580) [15]. 

In this paper, we report on the case of a 19-year-old girl with congenital myopathy and two heterozygous variants in the *CACNA1S* gene (c.1394-2A>G; c.1724T>C/p.L575P).

Given its recent identification and the few cases described in the literature with this phenotype, we describe the picture of a girl with a peculiar clinical expressiveness, expanding upon the phenotype of *CACNA1S* gene mutation, in addition to a systematic review on this disease that hopes to promote greater knowledge of it in order also to support diagnostic and therapeutic advances.

## 2. Materials and Methods

### 2.1. Search Strategy and Selection Criteria

A systematic search strategy was conducted following the guidelines of Preferred Reporting Items for Systematic Reviews and Meta-Analyses (PRISMA) [16].

The literature search was carried out on three electronic databases: PubMed (https://pubmed.ncbi.nlm.nih.gov/ accessed on 3 December 2022), Scopus (https://www.scopus.com/ accessed on 3 December 2022), and EBSCO (www.ebsco.com accessed on 3 December 2022). The search strategy, designed to include all fields (such as titles, abstracts, keywords, and all tests), used strings adapted for each database: (“*CACNA1S*” AND “congenital myopathy”). We carefully searched the reference lists of the original studies and reviews identified by the search, in order to identify additional studies meeting our criteria. We used the research tool “Zotero” (https://www.zotero.org/download/ (accessed on 3 December 2022)) to collect all the results in a single library.

The criteria used to select the articles were patients with congenital myopathy and a homozygous or heterozygous mutation in *CACNA1S* gene.

The exclusion criteria were: (i) duplicates; (ii) studies not concerning the aim of the paper; (iii) articles written in languages other than English, French, or Italian; (iv) reports in the form of abstracts, reviews, theses and conference papers; and (v) studies that described patients with mutations in more genes than *CACNA1S* and therefore with a pathogenicity of variants in *CACNA1S* that was not certain [17].

The full text of all the potentially eligible articles and their supplementary information were obtained by two authors (G.M, M.S. and E.C.), working independently. Any disagreements were resolved through consensus or, when necessary, by a third reviewer (A.O.).

The PRISMA flowchart shows the process of identification and selection of the papers: 461 abstracts were initially retrieved; after removing duplicates, 367 records were screened, 356 of which were excluded because they were abstracts, reviews, meeting abstracts, or on a different topic, and 2 because the patients also had another variant in a different gene besides the one in *CACNA1S*. Finally, 9 studies were included in the systematic review (Figure 1).

### 2.2. Data Collection Process

Data on the genetic, clinical, muscle biopsy, and muscle imaging features of the patients reported in each of the included studies were collected.

We collected data on genetic mutations, symptom onset, motor development, muscle weakness, facial involvement, ophthalmoparesis, respiratory involvement, scoliosis, deep tendon reflex, cognitive delay, cardiac involvement, CK levels, muscle MRIs, and muscle biopsies for each patient (Table 1). If these were not listed in the description, we wrote Not Applicable (NA).

### 2.3. Genetic Analysis 

Each variant of the patients was studied to identify its position in the human *CACNA1S* gene sequence with Varsome (https://varsome.com/ accessed on 7 June 2023) [26] (Appendix A) and the protein structure with Uniprot (https://www.uniprot.org/ accessed on 7 June 2023) [27]. We report in Figure 2 the graphic representation of these data (Figure 2).

The UCSC genome browser (http://genome-euro.ucsc.edu accessed on 7 June 2023) was used to model the human *CACNA1S* gene sequence.

Each variant was also studied to assess the probability of pathogenicity with different bioinformatics systems, including Varsome (https://varsome.com/ accessed on 7 June 2023) [26], Combined Annotation De-pendent Depletion (CADD, https://cadd.gs.washington.edu/snv accessed on 7 June 2023) [28], and Clin Var (https://www.ncbi.nlm.nih.gov/clinvar/ accessed on 7 June 2023) (Appendix A). 

It has been shown that the optimal CADD phred-like score cutoff is between 20 and 25.

The mutations of our patient were detected by the IRCCS Stella Maris foundation laboratory using a multiexon amplicon panel containing a total of 241 genes known to be associated with muscular dystrophies and myopathies. This panel was realized using SureSelect technology (Agilent, Santa Clara, CA, USA) and SureDesign sofware (version 7.10.0.6 earray.chem.agilent.com/suredesign/ accessed on 7 June 2023). To analyze the data obtained from our study, we used a routine bioinformatic pipeline that adopts the QIAGEN Clinical Insight (QCI) Interpret analysis suite (Qiagen, apps.ingenuity.com accessed on 7 June 2023). To assign pathogenicity, we used the following criteria: a sequence quality score greater than 30, a read depth greater than 30, and a rare occurrence in publicly available polymorphic data sets (with a minor allele frequency of <0.01% for autosomal dominant and <0.1% for autosomal recessive genes) with less than 1 occurrence in the homozygosity in gnomADv2.1.1 (gnomad.broadinstitute.org/ accessed on 7 June 2023; macarthurlab.org/2018/10/17/gnomad-v2-1 accessed on 7 June 2023) [29].

### 2.4. Clinical Grading 

Three categories of severity (mild, moderate, and severe) were defined based on the clinical phenotypes of the patients reported. Muscle MRIs and muscle biopsies were considered but did not guide the choice. This was due to the unavailability of data for some patients and the variability in the interpretations of the report. 

### 2.5. Case Report

The girl was born at a 31-week gestational age from the first pregnancy of unrelated parents and was characterized by the finding of polyhydramnios and a reduced fetal growth in the third trimester. At birth, she presented global muscle hypotrophy, weak crying, and was slightly hyporeactive. Due to the presence of poor and irregular respiratory activity, she required mechanical ventilation for the first two days of life and thereafter non-invasive support until the 46th day of life, because of frequent episodes of desaturation and apneas. She was discharged at 37 weeks post-menstrual age in a good general condition, with a planned follow-up program for her prematurity.

Language developmental milestones were acquired regularly, while motor developmental milestones were acquired late; she walked without support at 19 months of age.

In her first three years of life, she presented frequent episodes of postprandial vomiting and inappetence, resulting in slowed growth curves.

At the age of 6, she was referred to pediatric endocrinologists for severe short stature. Growth hormone (GH) stimulation tests showed no GH deficiency. She started replacement therapy for her idiopathic short stature.

At the age of 7 years she began presenting muscle fatigue, mild proximal muscle weakness, ligamentous laxity, scoliosis, and difficulty with independent walking. A karyotype analysis and array-cgh analysis were performed, with the results being negative for the suspicion of dysmorphogenetic disease. At 12 years of age, she discontinued GH therapy due to the failure of stature growth.

At the age of 16, due to worsening muscular aspects, she was taken to the Pediatric Neurology Unit of the Santa Chiara University Hospital in Pisa and the Neuropsychiatry Unit of the IRCCS Stella Maris Institution to investigate suspected congenital myopathy.

In a neurological examination, she presented widespread muscular hypotrophy, ligamentous laxity, predominantly axial and cingular weakness, Gowers sign, and a rigid spine with left convex scoliosis. In addition, she showed difficulty with autonomous walking, especially for modest distances, with frequent falls, easy fatigability, and joint pain.

She also noted a myopia, mild sensorineural hearing loss, and a typical myopathic face with an ogival palate, nasal voice, and tagged micrognathia, as well as difficulties in chewing and swallowing.

The neurological component is also associated with psychiatric disorders. In fact, she presented with generalized anxiety disorder and a specific phobia regarding the ingestion of solid food due to swallowing difficulties, which required treatment with selective serotonin reuptake inhibitors (SSRIs).

Upon the suspicion of congenital myopathy, instrumental investigations such as a muscle biopsy were performed, showing a histopathological picture compatible with moderate primary muscle suffering. It showed a fiber size variability, occasional internal nuclei, and disorganization of the intermyofibrillar network (Figure 3).

Her nerve conduction velocity (NCV) was normal. An electromyography showed an increased percentage of polyphasic PUMs with a reduced shunt area on all the muscles examined, compatible with myopathic type involvement. Muscle magnetic resonance imaging (MRI) of the pelvis showed hypotrophy of the muscles and an adipose infiltration of the gluteus muscles and tensor fasciae latae muscles. A muscle MRI of her lower limbs showed hypotrophy of the thigh, with major involvement of the medio-posterior muscles and an adipose infiltration of the semitendinosus and sartorius muscles; the images of her legs also showed a mild adipose infiltration of the peroneus and postero tibialis muscles (Figure 4).

A brain MRI showed an enlargement of the IV ventricle and peri-cerebellar cerebrospinal fluid spaces, while the MRI of the medulla was normal.

A pulmonary evaluation with spirometry was performed, which showed a restrictive ventilatory pattern, and so was an echocardiogram, which revealed minimal mitral and pulmonary valve insufficiency. In addition, an audiometric examination showed sensorineural deafness. A TRIM37 and LMNA gene analysis was performed, with negative results.

Meanwhile, a next generation sequencing panel of the genes associated with myopathy revealed the presence of a heterozygous variant in the *CACNA1S* gene inherited from the father (c.1394-2A>G) and one in the same gene inherited from the mother (c.1724T>C/p.L575P ). C.1394-2A>G is a non-coding mutation and c.1724T>C is a missense mutation. The first one is defined as pathogenetic according to Varsome and it has a CADD score of 34.

The second one is defined as having an uncertain significance according to Varsome and Clin Var and it has a CADD score of 31 (Appendix A).

This allowed her clinical condition to be correlated with a genetic substrate compatible with congenital myopathy.

## 3. Results

### Descriptive Findings

Nine studies, reporting a total of 23 patients, were finally included in our systematic review. In the analyses, we also included the additional patient described in the present case report. Thus, the results reported hereafter refer to 24 patients (13 males (54%), 10 females (42%), and 1 unspecified (4%)). 

A total of 24 patients presented 23 different homozygous, heterozygous, and compound heterozygous mutations. In total, 23 mutations were identified in 17 different exons (1 mutation (4,4%) in exons 3, 5, 6, 9, 12, 20, 21, 24, 26, 30, 33, 35, 37, and 41, 2 mutations (8.5%) in exons 16 and 40, and 4 mutations (17%) in exon 18) and one intron (intron 10 (4.4%)). All the mutations had a Varsome index equal to or greater than three (“Uncertain significant”, “Likely pathogenetic”, or “Pathogenetic”). All the mutations had a CADD score above 20 (Appendix A).

The 23 mutations identified were also located in different parts of the α 1 subunit of DHPR. In total, 8/23 mutations were located in different transmembrane domains, 12/23 mutations were located in the cytosol, and 3/23 mutations were located in the endomysium. The intracytoplasmic tract between domains II and III (5/23) and the intracytoplasmic tract after domain IV (5/23) were the positions with the most mutations in our review (Figure 2).

In addition, 6/24 patients had cytoplasmic mutations after domain IV and 11/24 patients had cytoplasmic mutations between domains II and III (Appendix A).

The myopathy onset was antenatal/neonatal in 20 patients (83%), in early childhood in 3 patients (13%), and late in 1 (4%). Motor development was delayed in 12 patients (50%), normal in 2 patients (8%), and unspecified in 10 patients (42%), 5 of whom died prematurely. Muscle weakness was described in all the patients, with 18 presenting generalized weakness (75%, in 10 of which (56%) it was predominately axial and proximal), 3 with only proximal and axial (12.5%) weakness, and 3 being unspecified (12.5%). There was facial involvement in 18 patients (75%), with an arched palate in 14 (58%). Ophtalmoplegia was reported in nine cases (37.5%). Feeding problems recurred in 15 patients (62.5%), in particular abnormalities in swallowing and chewing, and 3 of these needed gastrostomy. Respiratory problems were described in 17 cases (70%), 9 of whom (37.5%) had mild involvement and 8 (33%) had severe involvement. Nine patients (37.5%) had scoliosis and ten were unknown. The deep tendon reflex was normal in 2 cases (8%), reduced in 1 case (4%), absent in 3 cases (12.5%), and not described in 18. A moderate cognitive delay was described in only one patient (4%), not delayed in two cases, and not specified in the rest. In our case, an echocardiography revealed minimal mitral and pulmonary valve insufficiency, which had not been described in the literature before. CK was normal in 15 cases (62.5%), elevated in only 2 cases (8%), and unknown in the rest.

Muscle MRIs were available for 14 patients (58%): 1 patient had no alteration; 1 patient had the involvement of only the gluteus maximum muscles; 3 patients had global muscle atrophy; 3 patients had a major involvement of the anterior thigh, 2 of which had a sparing of the anterior compartment in the leg; 1 patient had an atrophy of the high leg muscles and an adipose infiltration of the extensor muscles; 1 patient had global muscle atrophy and anterior thigh infiltration; and 3 patients had an anterior thigh atrophy, 1 of which had an adipose muscles replacement. Only our patient presented a global hypotrophy of the pelvis and lower limbs, with a posterior-anterior gradient of involvement in the thigh and an adipose infiltration of the gluteus, tensor fasciae latae, semitendinosus, sartorius, peroneus, and postero tibialis muscles.

Muscle biopsies were available for 18 patients (75%). All the patients had a fiber size variability that was suggestive of muscle distress. Five patients also had an alveolar aspect of the myofibrillar network and four patients had a disorganization of the intermyofibrillar network.

According to clinical phenotype, the patients were divided into three severity groups. A mild phenotype was found in 11/24 patients (46% patients 1, 2, 3, 4, 7, 8, 9, 11, 12, 20, and 23). In this group, the patients had normal or delayed motor development, only proximal and axial generalized muscle weakness, facial muscle involvement, none to slight swallowing problems, and none to mild respiratory involvement. Only 5/11 patients presented scoliosis. Only one patient (patient 20) had moderate respiratory involvement. He was still included in the mild phenotype group because he presented an adulthood onset of symptoms and had no further signs of severity (Figure 5). 

A severe phenotype was found in 8/24 patients (33% patients 6, 10, 13, 14, 15, 21, 22, and 24). In this group, the patients had delayed motor development, generalized muscle weakness, facial muscle involvement, slight to severe swallowing problems with gastrostomy, and severe respiratory involvement. Two out of the eight patients (patients 6 and 10) in this group were included because their muscle MRIs showed a marked atrophy, although they presented occasional swallowing problems (Figure 5).

A intermediate–moderate phenotype was found in 5/24 patients (21% patients 5, 16, 17, 18, and 19). Patient 5 had important swallowing issues. However, he was included in this group because his muscle MRI showed a mild involvement of the anterior tight and his respiratory function was normal. Patients 16 and 17 had severe respiratory involvement. However, they were included in this group because they had a slight muscle weakness (mild proximal and axial involvement and mild proximal in lower members) and swallowing problems. Patients 18 and 19 had important swallowing issues and a moderate involvement of muscle facies. However, they were included in this group because the muscle MRI was normal in patient 19 and the respiratory function was slightly impaired in patient 18 (Figure 5). 

We tried to make a genotypic–phenotypic correlation between the pattern of inheritance and location of mutations with the clinical phenotypes of the patients. Most patients with a heterozygous mutation showed a milder phenotype: 7/10 patients had a milder phenotype; 1 patient had a moderate phenotype; and 2 patients had severe phenotype. Patients with compound heterozygous mutations showed different grades of phenotype: four patients had a milder phenotype; four patients had a moderate phenotype; and two patients had a severe phenotype. All the patients with homozygous mutations had a severe phenotype. Patients with mutations between domains II and III (11/24) had heterogeneous phenotypes: four patients had a mild phenotype; two patients had a moderate phenotype; and five patients had a severe phenotype. Patients with mutations after domain IV (6/24) also had heterogeneous phenotypes: two patients had a mild phenotype; three patients had a moderate phenotype; and only one patient had a severe phenotype. 

## 4. Discussion

Recent innovations in genetic technologies combined with extensive studies on big families and well-defined cohorts have allowed for the identification of a considerable number of causative congenital myopathy genes, mainly coding for proteins associated with the sarcomere structure or its stability, which are, therefore, involved in the mechanism of contraction [6,30,31].

The *CACNA1S* gene encodes the pore-forming subunit of the voltage-dependent calcium channel (DHPR) that regulates the rapid and generalized release of calcium within myofibers to promote excitation–contraction coupling [7,18]. Both recessive and dominant mutations are linked with a reduction in protein levels and an impairment of the calcium release induced by depolarization, compromising excitation–contraction coupling with SR dilatation and myofibrillar disorganization [32,33,34].

The *CACNA1S* mutation is by now recognized as a cause of early-onset congenital myopathy, although the phenotypic expression of this mutation is still partially misunderstood.

The first report in literature concerning compound heterozygous *CACNA1S* pathogenic variants as potentially causing congenital myopathy was about a pediatric patient [20] and was described by Hunter and colleagues in 2015.

The patient described was hypotonic and critically weak at birth, never gaining the ability to considerably chew, suck, or swallow.

At his evaluation at 3 months of age, he presented mild micrognathia, dysphagia, oropharyngeal faintness, suggestive facial diplegia, and ophthalmoplegia. He had spontaneous movement of his upper and lower limbs, but his deep tendon reflexes were not evocable. A muscle biopsy and extensive histology demonstrated a modest myofiber and occasional internal nuclei and coarse myofibrillar architecture. He carried compound *CACNA1S* variants, each inherited from a heterozygous parent. Unfortunately, in this report, no functional authentication of the variant’s pathogenicity was specified, precluding a conclusion on the pathogenicity of the recognized variants.

A further possible association of *CACNA1S* mutations with congenital myopathy phenotypes was afterwards suggested in a study [18] including a cohort of 11 patients from 7 different families, with an age range from 8 to 60 years and a consistent phenotype of early-onset myopathy and mutations in the skeletal muscle α 1 subunit of DHPR, *CACNA1S*. Family histories revealed that two families had recessive inheritance, two had dominant transmission, and three were sporadic mutations. 

All patients presented congenital or early-onset hypotonia with progressive, generalized and predominately axial muscle weakness, facial involvement with or without ophthalmoplegia, elevated CK in one family, and episodes of periodic paralysis in one patient.

There were mild to severe respiratory affections and swallowing issues in most patients and scoliosis in six patients, but there was no cardiac involvement in any of them.

The morphological and ultrastructural studies made by the biopsies revealed centralized or internalized nuclei and focal zones of sarcomeric disorganization in numerous patients, as well as an “alveolar” aspect of the intermyofibrillar network, which is peculiar and could possibly be a histopathological hallmark for *CACNA1S* mutations. In addition, muscle imaging reiterated some patterns seen in other congenital myopathies [35,36]. 

Subsequently, between 2019 and 2021, 11 more patients were described. Seven patients had an autosomal recessive inheritance (three homozygous mutations and four compound heterozygous mutations), while four patients had an autosomal dominant inheritance [5,31,32,33,34,35,36]. 

Similar phenotypic aspects were recognized in a few other studies. Morales et al. [19] described two adult patients with a neonatal onset of severe hypotonia, feeding difficulties, and respiratory insufficiency, progressively worsening with age, with the development of proximal weakness and also distal hyperlaxity. In both of them, an AR transmission of a *CACNA1S* variant was confirmed, though a non-specific myopathic pattern with a type 1 fiber predominance in their muscle biopsies was underlined. 

A non-specific myopathic pattern in a biopsy was also found in an adult female with a *CACNA1S* mutation affected by a congenital myopathy [24]. She presented with delayed motor development and a neonatal onset of severe, generalized muscle weakness, prominently affecting the axial muscles.

Neonatal hypotonia with respiratory distress and swallowing issues was additionally reported by Francois-Heude et al. [21] in a male patient with a compound heterozygote inherited pathogenic variant in the *CACNA1S* gene. He also had ophthalmoplegia and amimia with generalized muscle weakness, which luckily showed a gradual and encouraging improvement at a 12-month evaluation. The muscle biopsy identified a centronuclear myopathy, while the MRI was normal.

The case of a member of an Italian family carrying a rare heterozygous and potentially pathogenic *CACNA1S* mutation was peculiar [22]. The 61-year-old woman had no history of neonatal or infant abnormalities, nevertheless, she progressively developed bilateral palpebral ptosis, dysphagia for liquid foods, symmetric muscular weakness (mostly proximal rather than distal), cramps, hypophonia, and obstructive sleep apnea by the age of 50. The muscle biopsy showed rare nuclear clumps, a fiber size variability, and a few type II angulated and grouped hypotrophic fibers with focal zones of myofibrillar disorganization, though the MRI underlined a diffuse hypotrophy, which was more prominent in the right deltoid muscle.

The clinical conditions described up to now show a picture of mild to moderate Dihydropyridine Receptor Congenital Myopathy. In the latter seven papers, more severe phenotypes and lethal early-onset disease were also described. 

Ravenscroft et al. [23] and Kausthubham et al. [24] narrated three cases with patients who died within the first month of life.

Lastly, a 2019 Turkish report [25] talked about a consanguineous family, whose three children all presented with severe early-onset hypotonia, muscle weakness, respiratory distress, swallowing dysfunction, ophthalmoplegia, and pes equinus deformity. All of them were taken to a neonatal intensive care unit because of respiratory insufficiency on the first day of life. Two of them died at three months of age due to respiratory failure, while child 3 resided in the neonatal intensive care unit for five months and was then discharged with a tracheostomy and percutaneous gastric feeding tube. The last investigation at five years of age showed motor delay with generalized and axial weakness, mild facial involvement with a high arched palate, total ophthalmoplegia, scoliosis, and pes equinus deformity.

Through the collection of all the patients described in the literature, our review has better defined the complex clinical phenotype of *CACNA1S*-related myopathy. This myopathy was caused by homozygous, compound heterozygous, or heterozygous mutations in the *CACNA1S* gene on chromosome 1q32. 

Our review also highlighted the wide heterogeneity within *CACNA1S*-related myopathies. This wide spectrum of clinical phenotypes ranges from more severe pictures with a fetal akinesia sequence causing early death to milder pictures with an onset in early childhood or adulthood. The characteristic features of most affected individuals were delayed motor development with generalized hypotonia, a weakness of the facial muscles with a high arched palate, progressive axial and limb muscle weakness with a proximal-distal gradient beginning soon after birth or in infancy, swallowing difficulties to the point of needing a gastrostomy in severe cases, and mild to severe respiratory involvement. 

Additional features may include external ophthalmoplegia, ptosis, and scoliosis. In the most severe forms, respiratory involvement can become important and lead to death. 

Muscle biopsies showed variable morphologic abnormalities, including a fiber size variability suggestive of muscle distress, alveolar changes in the intermyofibrillar network, and a focal disorganization of the intermyofibrillar network.

Muscle MRIs showed a variable gravity involvement of the muscles of the pelvis, thighs, and legs, with a characteristic anterior posterior gradient of hypotrophy and fatty infiltration of the thighs.

To sum up, our patient’s phenotypic expression was similar to the mild cases described in the literature. She had an antenatal history of polyhydramnios and a reduced fetal growth in the third trimester of pregnancy. She was born prematurely at 31 weeks of gestation, like the probands reported by Ravenscroft et al. At birth, she was weak, hypotonic, hypotrofic, and she developed moderate respiratory distress, requiring mechanical ventilation for the first two days of life and non-invasive support for the next two months, but she was then regularly discharged without any other need for respiratory support.

Similar to most other patients, she had delayed motor development and gradually matured a pervasive muscle weakness with prominent axial and proximal involvement, ligament hyperlaxity, stiffness, and a scoliotic attitude, as well as fatigue in autonomous walking, which still persists.

She also had a restrictive ventilatory pattern, without the need for any ventilatory support, and a loss of appetite with difficulty and fatigue in chewing and swallowing, for which she needed to consume mostly soft and semi-solid foods, but no percutaneous gastric tube has been necessary so far. She had a distinctive myopathic face with an ogival palate and tagged micrognathia, but no ophtalmoplegia was evaluated. Cardiac involvement was also described, like in the Turkish girl, but no pes equinus deformity was recognized. Different from the other patients, our girl additionally showed myopia, was hyposomic (<3SD according to Cacciari et al. Growth Charts 2006 [37]), and presented mild sensorineural hearing loss and psychiatric features.

The muscle MRI confirmed some aspects observed in patients with CACNA1S myopathy, such as hypotrophy and a fatty infiltration of the lower limb muscles. However, our patient presented a posterior-anterior impairment gradient in the thigh muscles, whereas the literature described a greater impairment of the anterior compartment of the thigh in these myopathies. The histological finding in the muscle biopsy was the same as that described in the literature.

A clear genotype–phenotype correlation cannot be made due to the few cases described. Most patients had mutations in the *CACNA1S* gene between domains II and III (11/24) and after domain IV (6/24). Most patients with a severe phenotype also had mutations between domains II and III (6/8). The possible association between variants between domains II and III and an often severe myopathic phenotype could be related to the function of the protein cytoplasmatic loop. The intracellular “loop” connecting domains II and III contains a critical region that interacts with the ryanodine receptor, therefore making it important for calcium release from the endoplasmic reticulum, which is necessary for muscle contraction.

## 5. Conclusions

In summary, our experience describes a case of CACNA1S congenital myopathy with some peculiar phenotypic aspects, expanding upon CACNA1S-related phenotypes and collecting a set of the few cases that have so far been reported in the literature, supporting the latest evidence for this new entity and laying the groundwork for better targeted diagnoses, genetic counseling, and therapeutic opportunities. Our review summarizes and assembles the various genotypic and phenotypic expressions of *CACNA1S* mutations recognized to date, hoping to improve the knowledge of it and stimulate further studies about this rare condition. However, further studies and data will be needed and collected to better clarify the diagnosis and molecular peculiarity of this Dihydropyridine Receptor Congenital Myopathy.

## Figures and Tables

**Figure 1 genes-14-01363-f001:**
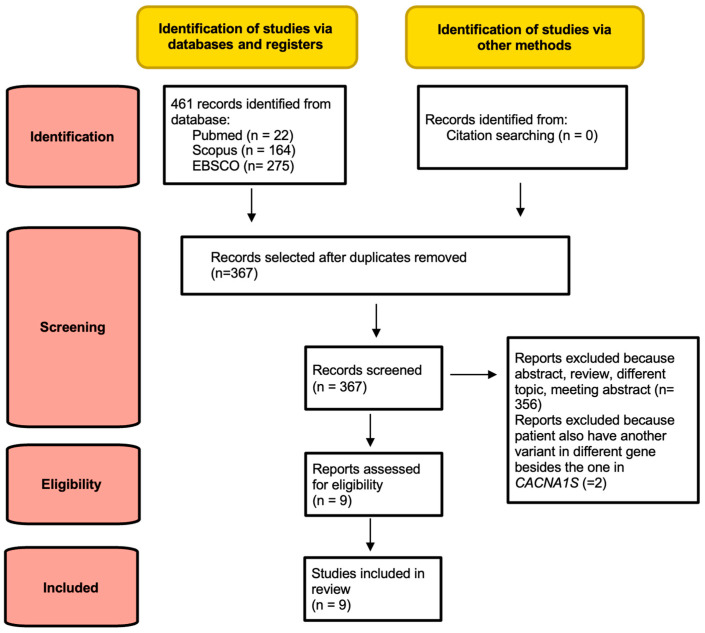
PRISMA flowchart of review.

**Figure 2 genes-14-01363-f002:**
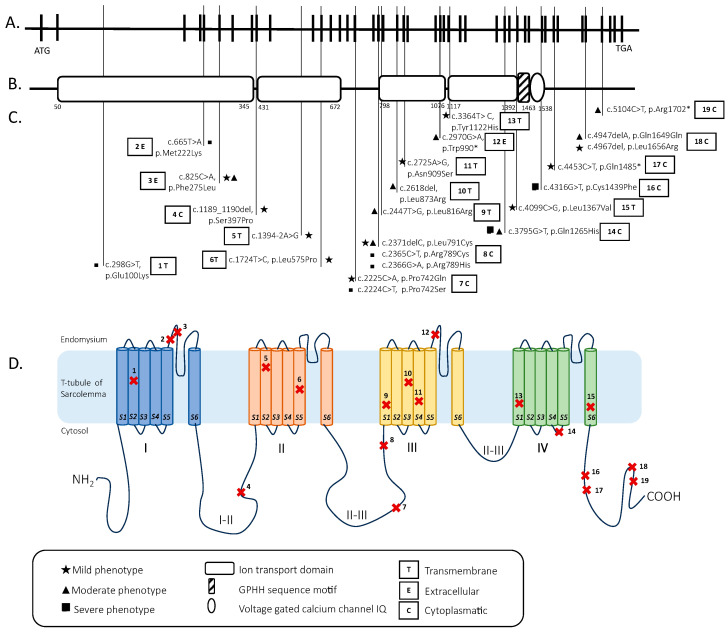
(**A**). The *CACNA1S* human gene extends over 73 kb and contains 44 exons (each vertical black line denotes one exon). The position of the start codon (ATG) is indicated. (**B**). The corresponding protein domains are represented according to the Protein Families (PFAM) database (**C**). Schematic view of the *CACNA1S* mutations of patients described in the present work with related grade (mild, moderate, and severe) of phenotype and position (transmembrane, extracellular, or cytoplasmatic) according to UNIPROT (https://www.uniprot.org/uniprotkb/Q13698/entry accessed on 7 June 2023) (**D**). Schematic view of the *CACNA1S*-encoded α 1 sub-unit of DHPR. The subunit has a total of four transmembrane domains (I–IV) composed by six segments (1–6) and three intracellular loop domains (loops I–II, loops II–III, and loops III–IV). Patient mutations are indicated with a cross and respective number.

**Figure 3 genes-14-01363-f003:**
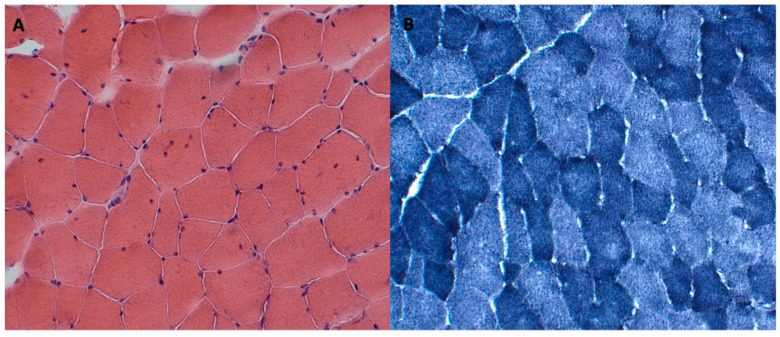
Left biceps brachialis muscle biopsy revealed fiber size variability, occasional internal nuclei, and disorganization of intermyofibrillar network (Hematoxylin-Eosin (**A**), NADH (**B**), stainings: 200×).

**Figure 4 genes-14-01363-f004:**
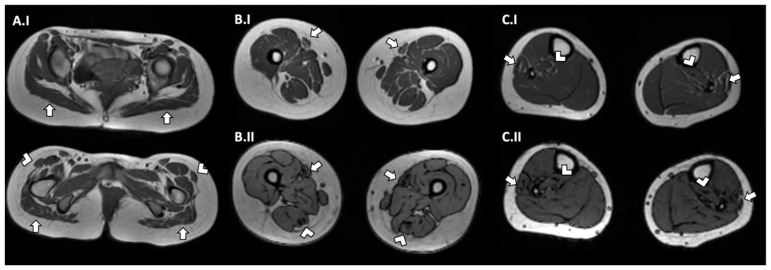
Muscle MRI images of axial T1 (I) and Ideal FSE (II) sequences of pelvis (**A**), thighs (**B**), and legs (**C**). MRI images showed: hypotrophy of pelvis muscles with adipose infiltration of gluteus muscles (A.I—white arrows) and tensor fasciae latae muscles (A.I—white arrow heads); hypotrophy of thigh with major involvement of medio-posterior muscles and an adipose infiltration of semitendinosus (B.I and B.II—white arrow heads) and sartorius muscles (B.I and B.II—white arrows); and mild adipose infiltration of peroneus (C.I and C.II—white arrow) and postero tibialis muscles (C.I and C.II—white arrow heads) in legs.

**Figure 5 genes-14-01363-f005:**
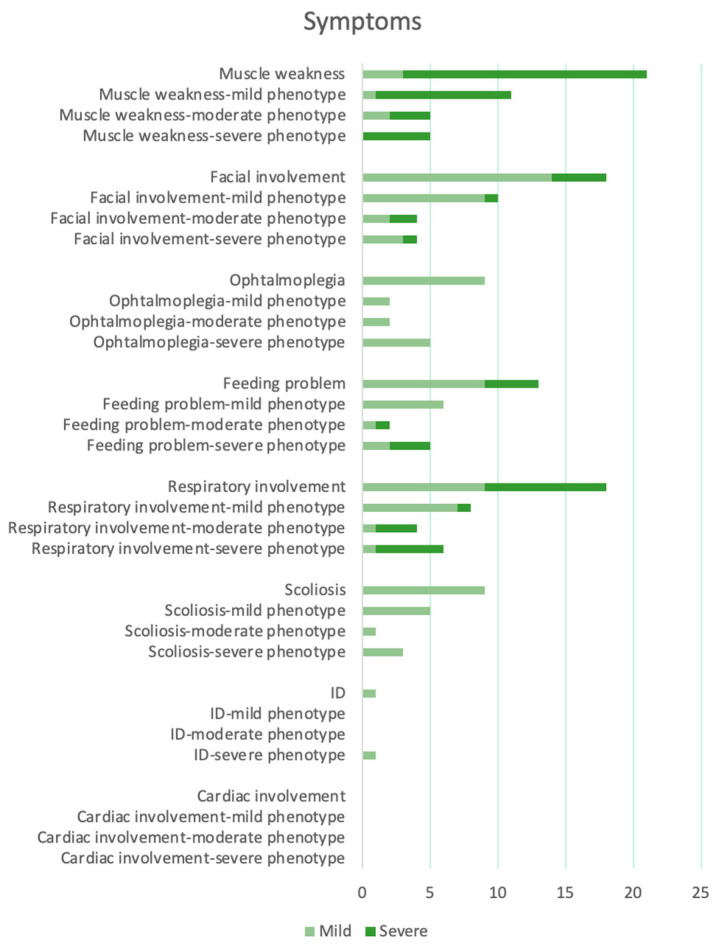
Graphic representation of symptoms in all patients and in different phenotypic groups. Muscle weakness: “Mild” indicates proximal and axial muscle weakness while “Severe” indicates generalized muscle weakness. Ophalmoplegia, Scoliosis, ID (Intellectual Disability): “Mild” indicates presence.

**Table 1 genes-14-01363-t001:** Description of phenotype and genotype of 24 selected patients NA: Non Applicable; P: Proximal; D: Distal; A: Anterior; and Devel.: Development.

	Number and Sex	Mutations	Onset	Motor Devel.	Muscle Weakness	Facial Involvement	Ophtalmo- Plegia	Feeding	Respiratory Inv.	Scoliosis	Deep Tendon Reflexes	ID	Cardiac Inv.	CK Level	Muscle MRI	Muscle Biopsy
Our case	1 F	c.1394-2A>G; c.1724T>C, p.L575P	Neonatal	Delayed	Generalized (A and P > D)	Moderate	No	Slight swallowing problem	Mild	Yes	Reduced	No	No	Normal	Hypotrophy of pelvis and thigh muscles (> medio-posterior). Adipose infiltration of gluteus, tensor fasciae latae semitendinosus, sartorius, peroneus, and postero tibialis muscles	Fiber size variability, occasional internal nuclei, and disorganization of intermyofibrillar network
Schartner et al. [18]	2 M	c.1189_1190del, p.Ser397Pro; c.4967del, p.Leu1656Arg	Antenatal/ neonatal	Delayed	Generalized (A and P > D)	Mild with high arched palate	Yes	Slight swallowing problem	Normal	No	NA	NA	No	Normal	Muscle atrophy	Centralized nuclei; focal disorganization; fiber size variability; alveolar aspect of the intermyofibrillar network; and uniformity of type I fibers
	3 M	c.4453C>T, p.Gln1485*; c.4967del, p.Leu1656Arg	Antenatal/ neonatal	Delayed	Proximal and axial	Mild with high arched palate	Yes	Slight swallowing problem	Mild	No	NA	NA	No	Normal	Fatty replacement and anterior muscle involvement in lower limb	Centralized nuclei; minicore; alveolar aspect of the inter-myofibrillar network; and uniformity of type I fibers
	4 F	c.825C>A, p.Phe275Leu; c.2371delC, p.Leu791Cysfs*37	Neonatal	Delayed	Generalized (P > D)	Mild with high arched palate	No	Normal	Normal	Yes	Normal	NA	No	Normal	NA	NA
	5 F	c.825C>A, p.Phe275Leu; c.2371delC, p.Leu791Cysfs*37	Neonatal	Delayed	Generalized (P > D)	Mild with high arched palate	No	Swallowing issues (G-tube)	Normal	Yes	Normal	NA	No	Normal	Mild involvement of the anterior thigh, predominantly in the vastus lateralis	NA
	6 M	c.298G>T, p.Glu100Lys; c.3795G>T, p.Gln1265Hisfs*57	Neonatal	Delayed	Generalized (A and P > D)	Mild with high arched palate and ptosis	Yes	Occasional swallowing difficulties	Severe	Yes	NA	NA	No	Normal	Marked atrophy involving all muscle groups of the upper leg bilaterally. Fatty infiltration, particularly of the extensor groups.	Fiber size variability; endomysial connective tissue around most fibers. Predominance of type I fibers
	7 F	c.2225C>A, p.Pro742Gln	Early childhood	Normal	Generalized	Mild with high arched palate	No	Normal	Mild	No	NA	NA	No	Normal	NA	Fiber size variability and alveolar aspect of the intermyofibrillar network
	8 M	c.2225C>A, p.Pro742Gln	Early childhood	Delayed	Generalized	Mild with high arched palate	No	Normal	Mild	yes	NA	NA	No	Normal	Muscle wasting of the anterior thigh and no fatty infiltration	Fiber size variability and alveolar aspect of the intermyofibrillar network
	9 M	c.2225C>A, p.Pro742Gln	Early childhood	Normal	Generalized	Mild with high arched palate	No	Normal	Mild	No	NA	NA	No	Normal	Muscle wasting of the anterior thigh and no fatty infiltration	Fiber size variability and alveolar aspect of the intermyofibrillar network
	10 M	c.2224C>T, p.Pro742Ser	Antenatal/neonatal	Delayed	Generalized	Mild with high arched palate	Yes	Occasional swallowing difficulties	Severe	Yes	NA	NA	No	Normal	Diffuse atrophy and more severe fatty infiltration in anterior compartment of the thigh and soleus and peroneal muscles in the leg	Rare internalized nuclei; fiber size variability; alveolar aspect of the inter-myofibrillar network; and uniformity of type I fibers
	11 M	c.4099C>G, p.Leu1367Val	Early onset (6–7 months)	Delayed	Generalized (more severe at upper member and P > D at lower members)	Mild with high arched palate	No	Occasional swallowing difficulties	Mild	Yes	NA	NA	No	Elevated (1000–2000)	Severe changes in thigh (anterior > posterior) and relative sparing of anterior compartment in the leg	Internalized nuclei; core-like structures; fiber size variability; and endomysial fibrosis
	12 F	c.4099C>G, p.Leu1367Val	Neonatal	Delayed	Generalized (more severe at upper member and P > D at lower members)	Mild with high arched palate	No	Occasional swallowing difficulties	Mild	Yes	NA	NA	No	Elevated (1000–2000)	Severe changes in thigh (anterior > posterior) and relative sparing of anterior compartment in the leg	NA
yis et al. [5]	13 M (died 3 m)	c.2366G>A, p.R789H *homo	Neonatal	NA	Generalized	NA	Yes	Absent suck	Severe	NA	NA	NA	No	NA	NA	Mild dystrophic changes such as contraction, regeneration, degeneration, nuclear internalization, and fibrosis were visible
	14 F	c.2366G>A, p.R789H *homo	Neonatal	Delayed	Generalized	Mild with high arched palate	Yes	Swallowing issues (Gastrostomy)	Severe	Yes	Absent	Yes	No	NA	NA	Marked variation in fiber size and shape; increased nuclear internalization; and grouping fascicules of large and small myofibers
	15 F (died 3 m)	c.2366G>A, p.R789H *homo	Neonatal	NA	Generalized	NA	Yes	Absent suck	Severe	NA	NA	NA	No	NA	NA	NA
Morales et al. [19]	16 F	c.2970G>A, p.Trp990*; c.5104C>T, p.Arg1702*	Neonatal	NA	Proximal and axial (mild)	NA	NA	Slight swallowing problem	Severe	NA	NA	NA	NA	Normal	Upper and lower limbs atrophy. No fat tissue replacement	Non-specific myopathic pattern with type 1 fiber predominance and mild myofibrillar disorganization
	17 M	c.2447T>G, p.(Leu816Arg)	Neonatal	NA	Proximal lower members	Mild with high arched palate	NA	Slight swallowing problem	Severe	NA	NA	NA	NA	Normal	Involvement of gluteus maximum	Non-specific myopathic pattern with type 1 fiber predominance
Hunter et al. [20]	18 M	c.4947delA, p.Gln1649Glnfs *72; c.3795G>T, p.Gln1265His	Neonatal	NA	Generalized (A and P > D)	Moderate with high arched palate	Yes	Swallowing issues (Gastrostomy)	Mild	NA	Absent	NA	NA	Normal	NA	Considerable myofiber size variation, polygonal small and large fibers, and occasional internal nuclei; and moderate architectural alterations in the form of coarse whorled fibers
Francois-Heude et al. [21]	19 M	c.2618del, p.(Leu873Argfs*21); c.5104C>T, p.(Arg1702*)	Neonatal	NA	Generalized	Moderate	Yes	Swallowing issues (Gastrostomy)	NA	NA	NA	No	NA	Normal	Normal	Centronuclear myopathy
Mauri et al. [22]	20 F	c.3364 T>C, p.Tyr1122His	Adult	NA	Generalized (P > D)	Mild with ptosis	NA	Slight swallowing problem	Moderate	NA	NA	NA	No	Normal	Diffuse hypotrophy, more prominent in the right deltoid muscle	Rare nuclear clumps, fiber size variability and few type II angulated and grouped hypotrophic fibers; and focal zones of myofibrillar disorganization
Ravenscroft et al. [23]	21 M (died 10d)	c.665T>A, p.Met222Lys; c.2365C>T, p.Arg789Cys	Antenatal/neonatal	NA	NA	NA	NA	NA	Severe	NA	NA	NA	NA	NA	NA	NA
	22 M (died 26wg)	c.665T>A, p.Met222Lys; c.2365C>T, p.Arg789Cys	Antenatal	NA	NA	NA	NA	NA	NA	NA	NA	NA	NA	NA	NA	Atrophy or myofibre disorganization
Grunseich C. et al. [24]	23 F	c.2725A>G, p.Asn909Ser	Neonatal	Delayed	Generalized (> A)	NA	NA	No	No	NA	NA	NA	No	NA	NA	Non-specific myopathy
Kausthubham et al. [25]	24 (died early)	c.4316G>T, p.Cys1439Phe	Neonatal	NA	NA	Cleft palate	NA	NA	NA	NA	NA	NA	NA	NA	NA	NA

## Data Availability

All procedures performed in this study were in accordance with the ethical standards of the institutional and/or national research committee and with the 1964 Helsinki declaration and its later amendments or comparable ethical standards.

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
