# Peer review of "Congenital Myopathy as a Phenotypic Expression of CACNA1S Gene Mutation: Case Report and Systematic Review of the Literature"

_genes, 2023, doi:10.3390/genes14071363_

Round 1

Reviewer 1 Report

The manuscript is devoted to a rare phenotype of congenital myopathy caused by variants of the CACNA1S gene.

The value of this work is due to the fact that the authors, in addition to describing their own observations, did an excellent job with the literature and made a review.

The main remark

1) The authors have made an excellent summary table. I believe that there is no need to repeat the information from it so thoroughly in the results and in the discussion.

I ask the authors to redo the discussion to reflect the possible causes of differences in the phenotypes of muscular dystrophy caused by this gene.

2)In addition, it seems to me it would be useful in principle to discuss the similarities and differences in the phenotypes of patients. Perhaps by making an additional table.This will facilitate the perception of the work and make it even more useful for doctors and researchers.

3)Materials and methods are clearly insufficiently described. For example, you need to specify the composition of the panel of genes, whether there were other findings.

4) It is also necessary to discuss the pathogenicity of the gene variants identified in the patient.

I consider this manuscript important, but I can recommend it for publication only after eliminating the comments and structuring the information.

Author Response

We want to thank the reviewer for the expert comments and the opportunity to improve our manuscript.

Concerning main remark:

1)The authors have made an excellent summary table. I believe that there is no need to repeat the information from it so thoroughly in the results and in the discussion. I ask the authors to redo the discussion to reflect the possible causes of differences in the phenotypes of muscular dystrophy caused by this gene.

Answer: We thank the expert referee for this comment. In the discussion of revised text, we summarized the phenotype of these patients by outlining all symptoms present, characteristic involvement on mucle MRI, and pattern of muscle biopsy (see lines 418-437) .

2)In addition, it seems to me it would be useful in principle to discuss the similarities and differences in the phenotypes of patients. Perhaps by making an additional table. This will facilitate the perception of the work and make it even more useful for doctors and researchers.

Answer: We appreciated this specific comment that offers the opportunity to clarify our study. In the results of revised text, we divided the patients into severity classes and we illustrated the symptoms of all patients and the differences and severity of symptoms for each severity class (see lines 282-313 and Figure 5) to better define the clinical phenotype.

3)Materials and methods are clearly insufficiently described. For example, you need to specify the composition of the panel of genes, whether there were other findings.

Answer: Thanks for this suggestion, we have better argued methods adding a "Genetic Analysis" section where the NGS panel used is also specified (see lines 112-154)

4) It is also necessary to discuss the pathogenicity of the gene variants identified in the patient.

Answer: Thanks for this suggestion, In the materials and methods of revised text, We have used different bioinformatics systems  to clarify the position and pathogenicity of variants. We have summarized this information in table S1 and in figure 2 (see lines 112-124 and 239-251). We also tried to make a genotypic-phenotypic correlation (see lines 316-327).

Reviewer 2 Report

In this review, the authors have summarized the phenotypes of 24 patients of congenital myopathy associated with mutations in Dihydropyridine Receptor (CACNA1S gene). The authors have selected only nine papers which includes 23 patients. They also did the phenotypes assessment of one patient which has shown also reduced fetal growth. Obviously, the phenotypes associated with congenital myopathy will show typical muscular dystrophy symptoms. The authors have summarized all of them. However, the quality of presentation is poor. Also the authors didnot try to analyze the associations between severity of disease and the position of mutation or the function of Dihydropyridine Receptor. My comments are provided below

1. The authors should describe about the structure of CACNA1S proteins and various mutations associated with it. 

2. The authors should provide a short tables for summarizing the result.

3. The authors have provided the details of mutation in CACNA1S but didnot provide any information about the type of mutations. They should provide the position of mutation in the protein i.e. is it in the transmembrane domain or intracellular. Did the authors find any information about protein level of CACNA1S.

4. The authors should discuss the phenotypes of these patients with other congenital myopathy  to show which phenotypes is more prominent due to CACNA1S mutation.

The English language needs improvement. The manuscript has many mistakes in the language.

Author Response

We want to thank the reviewer for the expert comments and the opportunity to improve our manuscript.

Concerning main remark:

1) The authors should describe about the structure of CACNA1S proteins and various mutations associated with it.

Answer: Thanks for this suggestion. In the introduction of text revised we have described the structure of CACNA1S protein (see lines 45-56). In materials and methods of text revised we have used different bioinformatics systems to clarify the position and pathogenicity of variants (see lines 112-124 and 239-251).

2) The authors should provide a short tables for summarizing the result.

Answer: We have summarized this information in Table S1 and in Figure 2.

3) The authors have provided the details of mutation in CACNA1S but did not provide any information about the type of mutations. They should provide the position of mutation in the protein i.e. is it in the transmembrane domain or intracellular. Did the authors find any information about protein level of CACNA1S.

Answer: We appreciated this specific comment that offers the opportunity to clarify positions and effects of mutations. In the results of text revised we described the position of mutations in CACNA1S gene and in CACNA1S protein (see lines 239-251) and we summarized these data in Table S1 and in Figure 2.

4) The authors should discuss the phenotypes of these patients with other congenital myopathy  to show which phenotypes is more prominent due to CACNA1S mutation.

Answer: We appreciated this specific comment that offers the opportunity to clarify our study. In the results of revised text, we divided the patients into severity classes and we illustrated the symptoms of all patients and the differences and severity of symptoms for each severity class (see lines 282-313 and Figure 5) to better define the clinical phenotype. We also tried to make a genotypic-phenotypic correlation (see lines 316-327).

Reviewer 3 Report

Dear Authors,

This is a well-organized review with a crucial focus the presentation of  congenital myopathy related with mutations in CACNA1S gene. Given the various phenotypic of CACNA1S related myopathy and the increasing number of mutations, it is of great importance to summarize and analyze common CACNA1S mutations for disease prevention and diagnosis.

There are no major problems with this review; the data are very well presented in text, tables and pictures.

Here are my comments and suggestions.

1.       Discussion: Are there therapeutic strategies for this type of congenital myopathy aimed at direct correction of the genetic defect, enzyme replacement therapy or pharmacological approaches.

2.       In general, symbols for genes are italicized. Consistency is needed in the presentation of gene names in the text.

Author Response

We want to thank the reviewer for the expert comments and the opportunity to improve our manuscript.

Concerning comments and suggestions:

1)Discussion: Are there therapeutic strategies for this type of congenital myopathy aimed at direct correction of the genetic defect, enzyme replacement therapy or pharmacological approaches.

Answer: We thank the expert referee for this comment. To date, there are no active clinical trials in men for this this type of congenital myopathy.

2) In general, symbols for genes are italicized. Consistency is needed in the presentation of gene names in the text.

Answer: Thank you for the note. We have revised the text to correct mistakes.

Round 2

Reviewer 1 Report

The authors made edits that significantly improved the article. Unclear points are explained. The work is structured and duplicates are removed. I have no comments, I consider the article worthy of publication.

Reviewer 2 Report

The authors have addressed all my concerns in the revised text. Now the quality of manuscript has significantly improved. I support the publication of the revised manuscript.

The authors have addressed all my concerns in the revised text. Now the quality of manuscript has significantly improved. I support the publication of the revised manuscript.